# Genetic Events Inhibiting Apoptosis in Diffuse Large B Cell Lymphoma

**DOI:** 10.3390/cancers13092167

**Published:** 2021-04-30

**Authors:** Etienne Leveille, Nathalie A. Johnson

**Affiliations:** 1Faculty of Medicine, McGill University, Montreal, QC H3G 2M1, Canada; etienne.leveille@mail.mcgill.ca; 2Department of Medicine, McGill University, Montreal, QC H4A 3J1, Canada; 3Departments of Medicine and Oncology, Jewish General Hospital, Montreal, QC H3T 1E2, Canada

**Keywords:** diffuse large B cell lymphoma, non-Hodgkin lymphoma, apoptosis, genetics, *BCL2*, NF-kB, *TP53*, mutations, translocations, amplifications

## Abstract

**Simple Summary:**

Diffuse large B cell lymphoma (DLBCL) is the most common type of non-Hodgkin lymphoma (NHL). Despite the genetic heterogeneity of the disease, most patients are initially treated with a combination of rituximab, cyclophosphamide, doxorubicin, vincristine, and prednisone (R-CHOP), but relapse occurs in ~50% of patients. One of the hallmarks of DLBCL is the occurrence of genetic events that inhibit apoptosis, which contributes to disease development and resistance to therapy. These events can affect the intrinsic or extrinsic apoptotic pathways, or their modulators. Understanding the factors that contribute to inhibition of apoptosis in DLBCL is crucial in order to be able to develop targeted therapies and improve outcomes, particularly in relapsed and refractory DLBCL (rrDLBCL). This review provides a description of the genetic events inhibiting apoptosis in DLBCL, their contribution to lymphomagenesis and chemoresistance, and their implication for the future of DLBCL therapy.

**Abstract:**

Diffuse large B cell lymphoma (DLBCL) is curable with chemoimmunotherapy in ~65% of patients. One of the hallmarks of the pathogenesis and resistance to therapy in DLBCL is inhibition of apoptosis, which allows malignant cells to survive and acquire further alterations. Inhibition of apoptosis can be the result of genetic events inhibiting the intrinsic or extrinsic apoptotic pathways, as well as their modulators, such as the inhibitor of apoptosis proteins, P53, and components of the NF-kB pathway. Mechanisms of dysregulation include upregulation of anti-apoptotic proteins and downregulation of pro-apoptotic proteins via point mutations, amplifications, deletions, translocations, and influences of other proteins. Understanding the factors contributing to resistance to apoptosis in DLBCL is crucial in order to be able to develop targeted therapies that could improve outcomes by restoring apoptosis in malignant cells. This review describes the genetic events inhibiting apoptosis in DLBCL, provides a perspective of their interactions in lymphomagenesis, and discusses their implication for the future of DLBCL therapy.

## 1. Introduction

Diffuse large B cell lymphoma (DLBCL) is the most common type of lymphoma and comprises about 25–40% of all non-Hodgkin lymphomas (NHL) in the Western world [1]. Patients are initially treated with chemoimmunotherapy, most commonly rituximab, cyclophosphamide, doxorubicin, vincristine, and prednisone (R-CHOP) [2,3]. Unfortunately, more than 50% of patients experience disease progression which, in young and fit patients, is treated with salvage chemotherapy followed by autologous stem cell transplant. Ultimately, only 10% of patients with relapsed or refractory DLBCL (rrDLBCL) are cured with this approach [4]. Work over the past decade has led to important insights into the disease biology. DLBCL is a heterogenous disease that evolves to evade apoptosis [5]. Many novel targeted therapies have been tested and are active in a subset of patients with rrDLBCL that is otherwise resistant to conventional chemotherapy [6]. Some of these have the potential to activate the intrinsic apoptotic pathway, independently of DNA-damage response pathways or P53, or engage the extrinsic apoptotic pathway through cell-mediated cytotoxicity. Furthermore, the genomic characterization of DLBCL at diagnosis and relapse allows for non-invasive monitoring of disease progression and clonal evolution over time using circulating tumor DNA (ctDNA) in the plasma [7,8]. This review focuses on the genomic alterations that affect critical survival and apoptotic pathways in DLBCL. An improved understanding of mechanisms that impair apoptosis in DLBCL may reveal vulnerabilities that could be exploited therapeutically in the future.

### 1.1. DLBCL Classification

DLBCL is classified into molecular subtypes that share common survival pathways and mechanisms that inhibit apoptosis. The most widely used classification stratifies DLBCL according to cell-of-origin (COO) molecular signatures, determined by gene expression profiling (GEP). These include the germinal center B cell-like (GCB) and activated B cell-like (ABC) subtypes, with 10–20% of cases having an intermediate or “unclassifiable” profile. GCB and ABC subtypes have distinct genomic alterations and clinical outcomes, with the latter being associated with an inferior overall survival [9,10,11,12]. The exact mechanisms of lymphomagenesis differ in GCB and ABC DLBCL, as they have distinct patterns of GEP and are, respectively derived from germinal center centroblasts and post-germinal center plasmablasts [13]. For instance, activation of the NF-kB pathway is mostly seen in ABC DLBCL, while *BCL2* translocations are almost exclusive to GCB DLBCL. Other genetic events, such as *TP53* mutations, are observed in both types of DLBCL [14]. An alternative gene expression classifier segregates DLBCL into three clusters that, for the purposes of this review, may have different mechanisms involved in survival and apoptosis. The OxPhos cluster is enriched in genes involved in mitochondrial function, oxidative phosphorylation, and the electron transport chain. The BCR/proliferation cluster expresses genes encoding components of the B cell receptor (BCR) signaling cascade, cell-cycle regulators, and DNA repair proteins. Finally, the host response (HR) cluster is characterized by increased expression of inflammatory mediators as well as components of the T cell receptor pathway and complement cascade [15]. The classification of DLCBL continues to evolve to include new information gained by recent whole exome or genome sequencing. The main genetic alterations identified in newer DLBCL classification systems are summarized in Table 1. These classification systems better represent the genetic heterogeneity of DLBCL and their distinct pathogenic pathways that could be modulated with targeted therapies. Although they overlap, significant differences are observed between those classification systems (reviewed in [16]), and a novel consensus classification remains to be determined [14,16,17].

### 1.2. Apoptosis in Normal Germinal Center B Cells

Apoptosis in the normal germinal center reaction is regulated by several pro- and anti-apoptotic proteins that have been previously reviewed [18]. In the normal maturation process of B cells in the germinal center, the intrinsic and extrinsic apoptotic pathways are essential to the proper functioning of positive and negative B cell selection [19]. The intrinsic pathway is triggered by cellular stress, including DNA alteration, that leads to mitochondrial permeabilization and release of pro-apoptotic factors, while the extrinsic pathway is triggered by ligands binding to death receptors and subsequent activation of the death-inducing signaling complex (DISC). Both pathways converge into the activation of effector caspases, the main apoptotic proteases (Figure 1) [20].

### 1.3. Inhibition of Apoptosis in DLBCL

A hallmark of cancer, including DLBCL, is the occurrence and accumulation of genetic alterations that promote malignant cell survival via inhibition of apoptosis [21]. This survival advantage can be exacerbated by the selective pressure imposed by therapies such as R-CHOP and is a major factor in the development of resistance to chemotherapy [22,23]. Accordingly, relapsed and refractory DLBCL (rrDLBCL) have a distinct genetic landscape that further inhibits apoptosis, contributing to resistance to additional chemotherapy and a poor outcome [4,7,24,25]. Understanding apoptotic pathways in DLBCL and the genetic factors contributing to their inhibition has the potential to dramatically improve the prognosis of DLBCL, notably by facilitating the development of targeted therapies that could selectively relieve apoptotic blockade and therefore restore response to treatment. Herein, the genetic events leading to inhibition of apoptosis in DLBCL are described (Table 2). The intrinsic and extrinsic apoptotic pathways, as well as their modulators, will be discussed.

## 2. Intrinsic Apoptotic Pathway

The intrinsic apoptotic pathway, also known as the mitochondrial or BCL2-regulated apoptotic pathway, is predominantly under the control of the BCL2 protein family [20,26]. These proteins are classified based on their influence on apoptosis and the number of BCL2 homology (BH) regions they contain. The anti-apoptotic proteins, which bind and sequester proapoptotic proteins, contain 4 BH domains (BH1–4) and include BCL2, MCL1, BCL-XL, and BCLW. The proapoptotic BH3-only proteins can be further divided into apoptotic sensitizers (BAD, NOXA, and HRK) that bind and inhibit anti-apoptotic proteins, and apoptotic activators (BID, BIM, and, to a lesser extent, PUMA) that can also directly activate the BH1–3 apoptotic effectors BAX and BAK [20]. Various cellular stresses, including oncogenes, DNA damage, and chemotherapy can trigger the intrinsic apoptotic pathway by activating the BH3-only proteins [27]. BAX and BAK are subsequently activated and permeabilize the outer mitochondrial membrane, allowing the release of cytochrome c in the cytosol, which is considered to be an irreversible commitment to apoptosis. Cytochrome c activates caspase 9, with APAF1 acting as a scaffold in the process. SMAC is also released from the mitochondrial outer membrane and leads to activation of caspase 9 by inhibiting XIAP, a caspase inhibitor. Activated caspase 9 in turn engages effector caspases, namely caspase 3, 6, and 7, leading to proteolysis and end stages of apoptosis (Figure 1) [20]. Inhibition of this pathway can be the result of alterations affecting different genes and proteins (Table 2) [28].

### 2.1. BCL2

BCL2 is the most common and important anti-apoptotic protein inhibiting apoptosis in DLBCL. *BCL2* is normally silenced in germinal center (GC) B cells to allow low affinity B cells generated through somatic hypermutation (SHM) to undergo apoptosis. Thus, its expression in DLBCL is pathogenic [19]. *BCL2* is located on chromosome 18q21 [29]. In addition to its 4 BH domains, the BCL2 protein structure is notable for the presence of a flexible loop domain that mediates interaction with the P53 tumor suppressor [30]. BCL2 protein expression in DLBCL can be caused by different genetic events such as translocations, mutations, gains, and amplifications as well as transcriptional upregulation from pathways discussed later (BCR signaling and NF-kB) (Table 2) [31]. *BCL2* translocations to immunoglobulin genes (IG), t (14;18) (q32; q21), t (2;18) (p11; q21) and t (18;22) (q21; q11), involving the *IgH*, *IgK* and *IgL* loci, respectively, are present in 20–25% of DLBCL, almost exclusively of the GCB subtype [32]. *BCL2* translocations are also a dominant feature in the new DLBCL classification systems, occurring in 71% of C3 DLBCL, 89% of BCL2 DLBCL, and 78% of EZB DLBCL [14,16,17]. They lead to constitutive transcription of *BCL2* by the enhancer elements within the active *IG* loci [33]. In addition to leading to *BCL2* overexpression, the t (14;18) translocation is associated with a significantly higher rate of SHM-associated *BCL2* mutations, which is the most commonly mutated gene in DLBCL [31]. The role of these mutations in the pathogenesis of DLBCL is unclear, as a large proportion are either synonymous or tend to occur outside of the functionally important BH domains [31]. Some of these mutations could still have a functional impact and a role in disease development. *BCL2* promoter mutations might increase BCL2 protein expression by preventing binding and transcriptional repression by BCL6 [34]. Mutations in the flexible loop domain might prevent P53 binding, resulting in increased sequestration of BAX by BCL2 and therefore reduced apoptosis [30]. Mutations in the BH4 domain of *BCL2* decrease calcium-mediated apoptosis by preventing binding of BCL2 to the inositol 1,4,5-triphosphate receptor (IP3R), a channel that promotes apoptosis by facilitating calcium conductance from the endoplasmic reticulum to the mitochondria [35,36]. However, taken collectively, *BCL2* mutations do not seem to affect prognosis [31]. *BCL2* gains and amplifications occur in up to 25% of DLBCL, mostly of the ABC subtype. They typically are the result of copy number aberration of chromosome 18q21.33 and are associated with increased BCL2 expression and worse prognosis [37].

### 2.2. MCL1

MCL1 is an anti-apoptotic protein from the BCL2 family that is essential in B cell development and germinal center formation. It is therefore normally expressed in GC B cells [38,39]. It directly binds and sequesters BAX and BAK and prevents their activation by BH3-only proteins [40]. MCL1 is deregulated in several types of lymphoma and contributes to lymphomagenesis in cell lines and mouse models [41,42,43]. In a recent study, strong MCL1 expression was predominantly seen in ABC DLBCL, and apoptosis was induced in MCL1-positive DLBCL cell lines after knockdown of MCL1 or treatment with the BH3 mimetic obatoclax [44]. In the same study, gains and amplifications of *MCL1* were observed in 26% of ABC DLBCL. However, *MCL1* mutations were identified in less than 1% of cases [44]. Other potential mechanisms for deregulation of MCL1 include upregulation by the proto-oncogene STAT3, which is expressed in ABC DLBCL [45], and decreased MCL1 degradation due to upregulation of the USP9X deubiquitinase [46].

### 2.3. BCLX and BCLW

*BCLX* produces two proteins via alternative splicing: BCL-XL, an anti-apoptotic protein, and BCL-XS, a BCL2 inhibitor [47]. BCL-XL is expressed in DLBCL, although its contribution to inhibition of apoptosis is unclear [44,48,49,50]. It remains a potential therapeutic target, as pharmacologic inhibition of BCL-XL leads to apoptosis in some DLBCL cell lines [44,49]. A major pitfall of BCL-XL inhibition is thrombocytopenia, as it is the primary survival factor in platelets [51]. *BCLW* is a relatively understudied anti-apoptotic gene, as it was initially shown to be only essential in spermatogenesis [52,53]. Overexpression of BCLW has been observed in DLBCL cell lines and has been associated with resistance to apoptosis and decreased patient survival [54,55]. However, these results have not been observed in a recent study [56]. Overall, there are no notable genetic alteration in *BCLX* and *BCLW* in DLBCL, and the role of the proteins encoded by these two genes is minor in comparison to *BCL2* and *MCL1* [44].

### 2.4. Pro-Apoptotic Proteins

Although genetic events amplifying the effect of anti-apoptotic proteins are well characterized, pro-apoptotic defects in DLBCL remain a relatively understudied phenomenon. Functional assessment of the intrinsic apoptotic pathway through BH3 profiling in DLBCL cell lines and primary samples has revealed defects in BH3-only pro-apoptotic proteins (class A apoptotic block) and in apoptotic effectors (class B block). Such pro-apoptotic defects are a factor contributing to resistance to the BCL2 inhibitor venetoclax [50,57]. However, they are unlikely to be the result of alterations at the genomic level, as mutations or other genetic events directly affecting pro-apoptotic proteins are rarely seen in DLBCL [24,28]. The exact causes of these pro-apoptotic defects remain to be determined, but could notably involve epigenetic silencing, transcriptional repression, or interaction with other proteins. There is also a possibility that genetic alteration of pro-apoptotic proteins is more common in rrDLBCL and a contributing factor to chemoresistance, as observed in other hematological malignancies [58,59,60].

In summary, deregulation of the intrinsic apoptotic pathway is a clear contributor to the pathogenesis of DLBCL. Translocations and mutations affecting *BCL2* are characteristic of GCB DLBCL, while *BCL2* and *MCL1* gains and amplifications are more common in the ABC subtype. These proteins can also be upregulated through other mechanisms that do not require direct alteration of the gene loci. The therapeutic relevance of these alterations is discussed at the end of this review.

## 3. Extrinsic Apoptotic Pathway

The extrinsic apoptotic pathway is important in the regulation of the germinal center reaction and prevention of lymphomagenesis [19]. It is also known as the death receptor-mediated pathway, as it is initiated by the binding of a death receptor ligand to its corresponding death receptor. These receptor-ligand pairs include FAS (CD95, APO-1) and FAS ligand (FasL), as well as TRAIL (APO2-L) and its receptors [61]. Activation of the death receptor leads to binding of its intracellular death domain to FADD (for FAS) or TRADD (for TRAIL receptors). The death effector domain of FADD/TRADD then binds to cFLIP, pro-caspase 8 (FLICE), and pro-caspase 10. The protein complex formed by FADD/TRADD, cFLIP, and pro-caspases is termed the death-inducing signaling complex (DISC) and allows conversion of pro-caspases into caspases. In the absence of death receptor ligand binding, formation of the DISC is inhibited by the anti-apoptotic regulator cFLIP [61]. The intrinsic and extrinsic apoptotic pathways converge when initiator caspases 8 and 10 activate effector caspases, resulting in proteolysis and apoptosis (Figure 1) [61]. Genetic events affecting proteins involved in different death receptor classes from the extrinsic apoptotic pathway have been reported in DLBCL (Table 2).

### 3.1. FAS Pathway

*FAS*, located on chromosome 10q23 and containing 9 exons, encodes the FAS cell surface death receptor [62,63]. The last exon of the gene encodes the death domain of the receptor, which is essential for initiation of FAS-mediated apoptosis [64]. *FAS* is highly expressed in germinal center B cells and is critical for the negative selection of suboptimal or self-reactive B cells during the germinal center reaction [65,66]. The mechanism by which FAS-mediated apoptosis is induced in the germinal center is not entirely understood, but might involve CD4+ T helper cells, which express FasL, and autonomous FAS-FasL signaling by B cells [19,67,68,69,70]. The FAS pathway is also one of the mechanisms by which cytotoxic T cells and natural killer (NK) cells kill their cellular targets, along with the perforin/granzyme pathway [71]. Deregulation of the extrinsic apoptotic pathway contributes to lymphomagenesis, notably by making NHL cells resistant to FAS-mediated apoptosis [72,73,74]. Heterozygous germline mutations in *FAS* are associated with autoimmune lymphoproliferative syndrome, a rare disorder characterized by lymphadenopathy, splenomegaly, autoimmune cytopenias, and a significantly increased risk of B cell lymphoma [75]. Somatic *FAS* mutations are consistently identified in GCB and ABC DLBCL, with reported frequencies of 5–15% [16,17,24]. The majority of mutations are located either in exon 9, encoding the death domain [64], or are frameshift or nonsense mutations that lead to loss of the death domain [76]. These mutations exert a dominant negative effect, as they prevent formation of the DISC and initiation of apoptosis via the extrinsic pathway [75]. Mutations are also seen in the 5′ region of the gene, which is also the case in healthy germinal center B cells, likely as a consequence of aberrant SHM [77]. In addition, *FAS* deletions have been observed in 7% of DLBCL [78]. Another potential mechanism leading to resistance to FAS-mediated apoptosis is an increased concentration of soluble FAS receptor (sFAS), a FasL sequestrant that results from alternative splicing out of exon 6 of *FAS* [79]. Prognostic information regarding *FAS* mutations in DLBCL is limited, but decreased FAS or FasL expression has been associated with decreased survival [80]. Preclinical studies with mouse models and lymphoma cell lines have explored different mechanisms of modulation of FAS-mediated apoptosis such as local administration of FasL, bispecific antibodies, and fusion proteins [81]. However, FAS-directed therapy is notably limited by severe hepatoxicity that precludes it from being used in clinical practice, and by the fact that FAS has other functions that are pro-oncogenic [82,83].

### 3.2. TRAIL Pathway

TRAIL (APO2-L) is a ligand from the tumor necrosis factor (TNF) family that can trigger extrinsic apoptosis by binding to either *TRAIL-R1* (DR4) or *TRAIL-R2* (DR5) [84,85]. The genes encoding these receptors, *TRAIL-R1* and *TRAIL-R2*, are both located on chromosome 8p21 [86,87]. Three other receptors can bind TRAIL and inhibit apoptosis by acting as decoys: TRAIL-R3 (DcR1), TRAIL-R4 (DcR2), and osteoprotegerin [84]. TRAIL-mediated apoptosis is one of the main effector mechanism of NK cells and is also used by cytotoxic T cells [88,89,90]. Highest levels of TRAIL receptors expression in B cells are seen in GC B cells, and evidence suggests that TRAIL-mediated apoptosis is an important regulator of B cell selection and germinal center homeostasis [91,92,93]. In addition, TRAIL and its receptors are important in immune surveillance against tumor development [88]. Resistance to TRAIL-mediated apoptosis has been reported in DLBCL [94]. A study that included 46 DLBCL has identified *TRAIL-R1* or *TRAIL-R2* mutations in 5 of them (10.9%), all of which were inside or in close proximity to the region encoding the death domain [95]. 8p21 deletions comprising *TRAIL-R1* and *TRAIL-R2* are also common in DLBCL [96]. TRAIL is an interesting therapeutic target, as it preferentially targets tumor cells and shows low levels of toxicity in animal models [84]. TRAIL agonists such as recombinant TRAIL, TRAIL-R antibodies, fusion proteins, and small molecules are being investigated preclinically in various malignancies, including DLBCL [97,98]. In addition, the TRAIL receptor agonist ABBV-621 is currently being investigated in a phase I clinical trial in solid tumors and hematological malignancies (NCT03082209; Appendix A).

## 4. Inhibitor of Apoptosis Proteins

The inhibitor of apoptosis proteins (IAPs) are a family of antiapoptotic proteins that inhibit the intrinsic and extrinsic apoptotic pathways, mainly via inhibition of caspases 3,7, and 9. Eight IAPs have been identified including XIAP, cIAP1, cIAP2, and survivin [99]. XIAP is a direct caspase inhibitor, while the other IAPs act by inhibiting SMAC, a XIAP inhibitor (Figure 1) [100]. Several of these IAPs can be upregulated in DLBCL [101,102,103,104]. As IAPs inhibit both the intrinsic and extrinsic apoptotic pathways due to their downstream action, they likely contribute to the pathogenesis of DLBCL and resistance to chemotherapy [99]. However, mutations in genes encoding IAPs are rare in DLBCL. Downregulation of IAPs is being investigated as a potential therapeutic strategy that could be combined with other apoptotic modulators in DLBCL [105,106,107].

## 5. P53 Pathway

*TP53*, located on chromosome 17p13.1 and containing 11 exons, encodes the tumor suppressor P53 and is the most frequently mutated gene in human cancers [108,109,110]. Germline mutations in *TP53* cause Li-Fraumeni syndrome, a cancer predisposition syndrome associated with breast cancer, brain tumors, adrenocortical carcinoma, leukemias, and many other malignancies [111]. P53 is a crucial regulator of cell cycle, cell proliferation, DNA repair, cellular senescence, and apoptosis [112]. Its structure is notable for a central DNA-binding domain that is necessary for the transcriptional activation of target genes. This domain contains several residues that are frequently mutated in different malignancies [112]. Under normal circumstances, MDM2 inhibits P53-mediated transcriptional activation by transporting P53 to the cytoplasm, binding its DNA-binding domain, and promoting its degradation. P53 also self-regulates in a negative feedback manner by inducing expression of MDM2 [113,114,115]. Cellular stresses such as DNA damage, oncogene activation, hypoxia, and loss of normal cell contact can lead to P53 activation by disrupting the binding of MDM2 to P53 [116]. Activated P53 can then exert its proapoptotic functions by modulating the transcription of several proteins. This results in an increased proportion of pro-apoptotic BCL2 proteins, increased expression of extrinsic pathway proteins FAS, FasL and TRAIL-R2, and upregulation of effector caspases 9 (via coactivator Apaf-1) and 6 (Figure 1) [117].

Genomic alterations in *TP53* and its associated proteins are common in DLBCL. *TP53* mutations are seen in more than 20% of GCB and ABC DLBCL and are associated with poor prognosis in the GCB subtype [118,119,120]. Approximately 90% of mutations lead to loss of P53 function, and most of them are located in the DNA-binding domain (exons 5-8), thus preventing P53-mediated transcriptional activation [119]. Mutant P53 can act as an oncogenic transcription factor and could therefore further contribute to lymphomagenesis if its expression is increased [121]. Chromosome 17p13.1 deletion occurs in approximately 10% of DLBCL, but does not seem to be correlated with survival [119]. The P53 inhibitor MDM2 is overexpressed in 40% of DLBCL, although gene amplifications are rare and expression is not affected by a common polymorphism (SNP309) [122]. *MDM4* and *RFWD2* encode two other P53 inhibitors and can both be amplified with gains of chromosome 1q23.3, reported in 15% of DLBCL [78]. Amplification of *BCL2L12*, an atypical BCL2 protein that inhibits P53 and caspases 3/7, have been observed in 10% of cases [78]. *CDKN2A* encodes the ARF (p14) and INK4a (p16) tumor suppressors [123]. ARF promotes activation of the P53 pathway by binding and inhibiting MDM2 [124]. *CDKN2A* deletions are present in 19–35% of DLBCL. They are associated with an ABC subtype and a decreased survival [14,125,126]. Alterations of the P53 pathway also include P53 target genes such as *PERP* (caspase 8 activator) and *SCOTIN* (pro-caspase 3/7 activator), which are deleted in 27% and 8% of DLBCL, respectively [78].

*TP53* is the most commonly mutated gene in rrDLBCL, with mutations observed in up to half of cases [7,24]. Clonal evolution studies have shown that most of these mutations are present in primary DLBCL subclones that are selected for during chemotherapy [7,127,128]. For this reason, tracking of *TP53* mutations in the plasma ctDNA in patients undergoing chemotherapy has the potential to detect and monitor early resistant clones [129,130].

## 6. Transcriptional Regulation of Apoptotic Pathways

In addition to alterations in the apoptotic pathways themselves, resistance to apoptosis in DLBCL is driven by the constitutive activation of transcriptional regulators that decrease apoptotic signals and potentiate anti-apoptotic proteins. Inhibition of apoptosis by the nuclear factor kappa beta (NF-kB) pathway is the most notable example of such transcriptional regulation in DLBCL [131]. The role of the NF-kB pathway in DLBCL and other hematological malignancies has been previously reviewed [131]. In B cells, activation of the NF-kB pathway is triggered by B cell receptor (BCR) signaling. This leads to formation of the CARD11-BCL10-MALT1 (CBM) complex, which allows translocation of cytoplasmic NF-kB to the nucleus to facilitate target gene transcription [132,133,134]. Regulation of apoptosis by the NF-kB pathway affects both the intrinsic and extrinsic apoptotic pathways by targeting and upregulating cFLIP [135,136], BCL2 [137], BCL-XL [138], cIAP1 [139,140], cIAP2 [139,140], XIAP [141], and survivin [142] (Figure 1). Constitutive NF-kB activation is a hallmark of ABC DLBCL and can involve different components of the pathway (Table 2) [143].

### 6.1. B Cell Receptor Signaling

In ABC DLBCL, the NF-kB pathway is constitutively activated by chronic active BCR signaling [144]. This involves BCR signal transduction via phosphorylation of the immunoreceptor tyrosine-based activation motifs (ITAMs) of CD79A and CD79B. Phosphorylated ITAMs trigger a signaling cascade that sequentially activate the SYK and BTK tyrosine kinases and results in activation of the CBM complex [144,145]. Gain-of-function mutations in the ITAM of *CD79B* occur in 10–25% of ABC DLBCL and 3% of GCB DLBCL, and lead to increased BCR signaling [14,144,146]. *CD79A* mutations can also be observed but are rare [144]. Potential loss-of-function mutations in genes encoding negative regulators of BCR signaling, such as *LYN*, *LAPTM5*, *PTPN6*, *GRB2*, *PRKCD*, *DGKZ*, *SLA*, and *MAP4K1* have collectively been identified in almost 40% of all DLBCL [14]. Of note, in addition to NF-kB, BCR signaling upregulates several pathways that contribute to cellular proliferation and oncogenesis, notably the MEK/ERK and AKT/PI3K pathways [5].

### 6.2. CARD11-BCL10-MALT1 Complex

As described above, the CBM complex is essential for NF-kB activation in lymphocytes. In resting cells, CARD11, a scaffold protein from the membrane-associated guanylate kinase family, is inactivated by its autoinhibitory domain. B cell activation leads to phosphorylation and activation of CARD11, recruitment of BCL10 and MALT1, formation of the CBM complex, and NF-kB activation [147,148]. Missense mutations in *CARD11* occur in approximately 10% of ABC DLBCL and in a smaller proportion of GCB DLBCL. These mutations occur almost exclusively in the coiled-coil domain of *CARD11* and result in a gain of function by preventing functioning of the autoinhibitory domain [147]. Chromosomal rearrangements involving *BCL10* are identified in up to 20% of DLBCL and are more common in the GCB subtype [149,150]. *BCL10* amplifications can also occur but are rare [14]. In addition to its role in the NF-kB pathway, wild-type *BCL10* also has pro-apoptotic functions [151]. *BCL10* mutations occur in about 5% of DLBCL [14,17] and result in a loss of BCL10 pro-apoptotic function while preserving its ability to activate NF-kB [151]. As for *BCL2*, *MALT1* is located on chromosome 18q21 and can therefore be gained in the same copy number alteration events [152].

### 6.3. NF-kB Genes and Regulators

NF-kB is a family of five dimeric transcription factors (RelA, RelB, c-Rel, p50, and p52) that are involved in numerous processes such as cellular development and proliferation, immune cell activation, and regulation of apoptosis. Two other proteins, p100 and p105, are precursors of p52 and p50, respectively [133]. These 7 proteins share a Rel Homology Domain (RHD) that mediates their dimerization, interactions with inhibitors, and DNA binding [133]. Despite the prominent role of the NF-kB pathway in ABC DLBCL, mutations affecting the NF-kB genes themselves are rare [153]. Other rare genetic events directly involving NF-kB include chromosome 10q24 rearrangements that lead to loss of the 3′-end of *NFKB2* (encoding p100) and constitutive protein activation [154], and amplifications of *REL* (encoding c-Rel), which are mostly seen in GCB DLBCL [9]. However, the significance of *REL* amplifications is unclear, as they do not correlate with NF-kB target gene expression [154]. Mutations involving positive and negative NF-kB regulators are more common and are collectively seen in more than half of ABC DLBCL, and in more than 20% of GCB DLBCL [153]. *A20* is located on chromosome 6q23.3 and encodes a ubiquitin-modifying enzyme that can downregulate the NF-kB response [155]. Loss-of-function mutations in *A20* have been identified in 24% of ABC DLBCL [153]. Inactivation of *A20* can also be the result of 6q23 deletions [156,157]. Mutations in *TNIP1*, encoding an A20-binding inhibitor of NF-kB, can also be seen [14]. *MYD88* encodes an adaptor protein that upregulates the NF-kB pathway by mediating toll and interleukin-1 signaling. The *MYD88^L265P^* mutation, a gain-of-function mutation that leads to constitutive NF-kB activation, is observed in 18–30% of ABC DLBCL and is associated with poor prognosis [14,16,158]. In addition, this mutation is particularly prevalent in primary extranodal DLBCL and is notably present in most cases of primary testicular lymphoma and primary central nervous system lymphoma [14,16,17].

Inhibition of apoptosis is the main mechanism by which NF-kB contributes to the pathogenesis of DLBCL, particularly of the ABC subtype [131]. This anti-apoptotic effect affects the intrinsic and extrinsic apoptotic pathways as well as the inhibitor of apoptosis proteins (Figure 1, Table 2). Numerous steps of the NF-kB can be altered, which may lead to different responses in attempts to pharmacologically downregulate this pathway.

## 7. Therapeutic Targeting of Apoptosis

The central role of inhibition of apoptosis in DLBCL and other malignancies has led to increasing investigation of apoptotic pathways and proteins as potential therapeutic targets [159]. Venetoclax is a BH3 mimetic and BCL2 inhibitor studied in multiple hematological malignancies. It is approved in chronic lymphocytic leukemia (CLL) and in older acute myeloid leukemia (AML) patients deemed unfit for conventional chemotherapy [160]. Initial trials in DLBCL have demonstrated modest activity [161,162,163], but a recent phase II study adding venetoclax to R-CHOP for previously untreated DLBCL has shown promising results, particularly in BCL2-positive cases [164]. One of the main factors contributing to venetoclax resistance in DLBCL is upregulation of MCL1, and inhibition of this protein can induce apoptosis and increase venetoclax-induced apoptosis in DLBCL cell lines [41,50,165,166,167]. DLBCL where resistance to apoptosis is driven by an increase in anti-apoptotic proteins (class C apoptotic block) might therefore respond to venetoclax with the possible addition of an MCL1 inhibitor. Doxorubicin and vincristine might also be beneficial in such patients, as these agents are associated with decreased levels of MCL1 and increased venetoclax-induced apoptosis [50]. Another factor contributing to inhibition of intrinsic apoptosis is pro-apoptotic functional defects (class A and B blocks). DLBCL with such defects might benefit from therapies that trigger cell death independently of the mitochondrial apoptotic pathway, for example by inducing cell-mediated cytotoxicity [57].

Extrinsic apoptosis is of particular importance in the use of therapies that involve cytotoxic T cells, namely immune checkpoint inhibitors, chimeric antigen receptor T (CAR T) cells, and bispecific T cell engagers (BiTEs) [168,169,170]. Immune checkpoint inhibitors have not been studied extensively in DLBCL, but have shown relatively low response rates in rrDLBCL [171,172]. This could be explained by the fact that most DLBCL are not characterized by robust T cell infiltration and activation. However, the immune landscape of DLBCL is heterogeneous and certain subtypes, such as those with constitutive NF-kB activation, could still benefit from immune checkpoint inhibition [173]. Anti-CD19 CAR T cell therapy has shown good response rates in selected cases of rrDLBCL and is approved as third-line therapy in multiple countries [174]. Blinatumomab, a BiTE that links CD3-positive T cells and CD19-positive B cells, has shown a 43% overall response rate and a 19% complete remission rate in a phase 2 trial including 25 patients with rrDLBCL [175]. It can be hypothesized that response to these therapies requires a functional extrinsic apoptotic pathway, which could explain the lack of response in a significant proportion of patients. Recent evidence using CRISPR knockout screens suggests that defects in the extrinsic apoptotic pathway contribute to resistance to CAR T cell and BiTE therapy in DLBCL and other hematological malignancies [176,177]. Such defects seem to exert a dominant negative effect and also prevent perforin/granzyme cytotoxicity by leading to prolonged antigen exposure and subsequent T cell exhaustion and dysfunction [177]. However, the nature of the extrinsic apoptosis defects contributing to T cell-based therapy resistance remains to be characterized. This has several potential clinical implications: for instance, DLBCL with mutations in genes that encode components of the extrinsic apoptotic pathway might have suboptimal responses or higher rates of resistance to CAR T cell or BiTE therapy. Patients receiving these therapies might also benefit from combination therapy that simultaneously targets the intrinsic apoptotic pathway, as shown in B cell malignancies cell lines where CAR T cell therapy was combined with the BH3 mimetic ABT-737 [178]. Currently, therapies involving cytotoxic T cells remain mostly used and investigated in rrDLBCL (Appendix A).

An important consideration in the choice of therapy for rrDLBCL is a high rate of *TP53* mutations [7]. Loss of P53 function notably drives the development of chemoresistance by blunting the DNA-damage response and downregulating the initiation of apoptosis [179]. This implies that agents depending on P53 to trigger apoptosis by inducing DNA damage, such as doxorubicin, are unlikely to be effective in a large proportion of rrDLBCL [179]. BH3 mimetics, T cell-based therapies, IAPs inhibitors, and other therapeutic strategies that kill cells independently of P53 might therefore be more beneficial in those cases. Pharmacologic inhibition of the P53 inhibitor MDM2 has also shown potential in preclinical studies [180]. However, two recent phase 1 trials combining the MDM2 inhibitor idasanutlin with rituximab (plus venetoclax in one of the two trials) in rrDLBCL were terminated because of the overall modest benefits observed (NCT03135262 and NCT02624986).

Another potential therapeutic strategy in DLBCL is to modulate transcriptional regulators of apoptosis. The predominant effect of NF-kB on inhibition of apoptosis in DLBCL, particularly of the ABC subtype, makes it a therapeutic target of interest. Inhibition of BCR signaling with the BTK tyrosine kinase inhibitor ibrutinib has initially shown good tolerability but modest response rates in ABC DLBCL [181,182]. Phase 1 trials using ibrutinib in combination with either lenalidomide or R-ICE (rituximab, ifosfamide, carboplatin, and etoposide) have shown good response rates in non-GCB rrDLBCL [183,184]. However, the addition of ibrutinib to R-CHOP did not improve progression-free or overall survival in patients with de novo non-GCB DLBCL [185]. Lenalidomide is an immunomodulatory agent that downregulates BCR signaling [186]. Its addition to R-CHOP in *de novo* ABC DLBCL has shown promising results in phase II trials [187,188], but a recent phase III trial has shown no improvement in progression-free or overall survival in previously untreated ABC DLBCL [189]. Bortezomib, a proteasome inhibitor that downregulates NF-kB by decreasing the degradation of inhibitory kB proteins [190], has shown no benefit in phase II and III trials when combined with R-CHOP in *de novo* ABC DLBCL [191,192,193]. The disappointing results seen with modulation of NF-kB do not exclude this pathway as a potential therapeutic target, as the events leading to NF-kB upregulation in DLBCL are heterogeneous. For instance, it can be hypothesized that BTK inhibition with ibrutinib could have a limited effect in a DLBCL with a mutation that affects a downstream step in the NF-kB pathway. Modulation of the NF-kB pathway could therefore be of therapeutic benefit in selected patients, particularly if used in conjunction with other apoptotic modulators that are chosen based on apoptotic profiling. The new DLBCL classification systems might contribute to better identification of patients that would best respond to NF-kB downregulation or other therapies that aim to restore apoptosis. Therapeutic targeting of apoptosis is currently being investigated in newly diagnosed DLBCL and rrDLBCL in numerous clinical trials (Table 3; Appendix A).

## 8. Conclusions

DLBCL genetically reprograms itself to inhibit apoptosis. This inhibition of apoptosis is a consequence of genetic events leading to dysregulation of several interacting pathways that modulate intrinsic and extrinsic apoptosis (Table 2). Overall, this results in an increase in anti-apoptotic BCL2 proteins, dysfunction of death receptor and DISC signaling, and blunting of the P53-mediated DNA-damage response. In addition, this anti-apoptotic state is upregulated by BCR signaling and the NF-kB pathway (Figure 1). Ultimately, inhibition of apoptosis confers a survival advantage to malignant cells, especially under the selective pressure of chemotherapy. As a result, inhibition of apoptosis is even more predominant in rrDLBCL, which harbor very high rates of *TP53* mutations [7,8,194,195].

In addition to improving the understanding of the pathogenesis of DLBCL, characterizing the genetic events inhibiting apoptosis have several important clinical implications. Indeed, newer DLBCL classification systems, which are based on genetic defects, recapitulate the different mechanisms of inhibition of apoptosis more accurately than the COO classification. Once a consensus classification system is established, it would likely replace the current COO classification in genetic studies and clinical trials. Sequencing lymphomas in clinical practice may reveal mutation patterns that would help in establishing the diagnosis of DLBCL. Monitoring these mutations in the plasma ctDNA over time would be a non-invasive approach to identify the emergence of resistant subclones. This could then lead to the timely implementation of targeted therapies. For instance, ibrutinib could be beneficial in MCD and BN2 DLBCL, which are characterized by predominant B cell receptor-dependent NF-kB activation, while BH3 mimetics might be particularly beneficial in the EZB subtype, which shows frequent *BCL2* translocations [14,16,17].

A better understanding of the genetic events contributing to the inhibition of apoptosis in DLBCL is essential. It gives insight into the mechanisms involved in disease development and progression. The identification of potential therapeutic targets in the plasma could help guide physicians to select effective therapies based on DLBCL subtype or apoptotic profile. This precision medicine may ultimately improve the survival of patients with rrDLBCL.

## Figures and Tables

**Figure 1 cancers-13-02167-f001:**
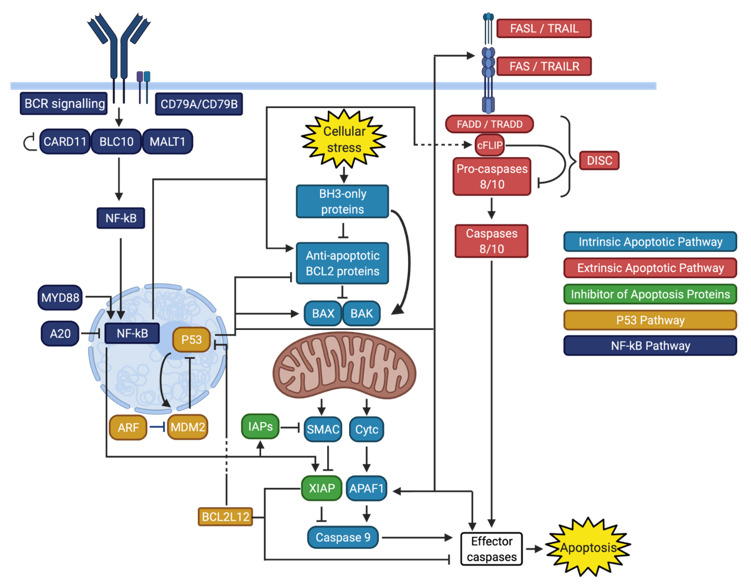
Simplified Overview of Apoptosis in DLBCL, BCR: B cell receptor, Cytc: cytochrome C, DISC: death-inducing signaling complex, IAPs: inhibitor of apoptosis proteins. Created with BioRender.com.

**Table 1 cancers-13-02167-t001:** Overview of Recently Proposed Genetic Classifications of DLBCL.

Lacy et al. [16]	Chapuy et al. [17]	Schmitz et al. [14]	Predominant COO Subtype
**MYD88 (16%)** MYD88^L265P^, *CD79B*, *PIM1*, and *ETV6* mutations 9p21.3/*CDKN2A* deletions	**C5 (21%)** 18q gains *CD79B* and *MYD88^L265P^* mutations	**MCD (8%)** *MYD88^L265P^* and *CD79B* mutations	**ABC DLBCL**
**BCL2 (19%)** *BCL2* mutations and translocations *EZH2*, *CREBBP*, *TNFRSF14*, *KMT2D*, and *MEF2B* mutations	**C3 (18%)** *BCL2* mutations and translocations *PTEN* inactivation Mutations in chromatin modifiers Alterations in BCR and PI3K signaling	**EZB (22%)** *EZH2* mutations and *BCL2* translocations	**GCB DLBCL**
**SOCS1/SGK1 (12%)** *SOCS1, SGK1, CD83 NFKBIA, HIST1H1E*, and *STAT3* mutations	**C4 (17%)** Mutations in NF-kB modifiers, immune evasion molecules, core histone genes, and RAS/JAK/STAT pathway components		**DLBCL NOS**
**TET2/SGK1 (11%)** *TET2, SGK1, KLHL6, ZFP36L1, BRAF, MAP2K1,* and *KRAS* mutations			**GCB DLBCL**
**NOTCH2 (15%)** *BCL6* rearrangements *NOTCH2*, *BCLL10, TNFAIP3, CCND3, SPEN, TMEM30A, FAS,* and *CD70* mutations	**C1 (19%)** *BCL6* rearrangements *MYD88^non-L265P^*, *FAS*, NOTCH2 pathway, and NF-kB pathway mutations	**BN2 (15%)** *BCL6* fusions and *NOTCH2* mutations	**GCB DLBCL, ABC DLBCL, and DLBCL NOS**
**NEC (26%)** Not elsewhere classified		**Other (54%)**	
	**C2 (21%)** *TP53* mutations 17p/*TP53*, 9p21.3/*CDKN2A* and 13q14.2/*RB1* deletions		**ABC DLBCL and GCB DLBCL**
	**C0 (4%)** No defining genetic driver		
		**N1 (2%)** *NOTCH1* mutations	**ABC DLBCL**

Percentage of cases reported is indicated next to each subtype. Reprinted with permission from ref. [16]. 2020 Elsevier.

**Table 2 cancers-13-02167-t002:** Main Genetic Events Inhibiting Apoptosis in DLBCL.

Intrinsic Apoptotic Pathway	Role in Apoptosis	Genetic Alterations
*BCL2*	Inhibition of intrinsic apoptotic pathway via inhibition of pro-apoptotic BCL2 proteins	Increased expression: translocations (mainly t(14;18)), gains and amplifications (18q21.33), mutations (promoter region, BH4 domain, and flexible loop domain)^14,16,17, 31−33, 37^
*MCL1*		Increased expression: gains and amplification^43,44^
**Extrinsic apoptotic pathway**		
*FAS*	Induction of FAS-mediated apoptosis after binding of FasL	Decreased expression: mutations (death domain), deletions^16,17,24,64,76−78^
*TRAIL-R1*/*TRAIL-R2*	Induction of TRAIL-mediated apoptosis after binding of TRAIL	Decreased expression: mutations (death domain), deletions (8p21)^95−96^
*CASP10*	Effector caspases activator	Decreased expression: inactivating mutations^17^
**P53 pathway**	Promotion of intrinsic and extrinsic apoptosis via upregulation of several pro-apoptotic proteins	
*TP53*		Decreased expression: mutations (DNA-binding domain), 17p13.1 deletions; polymorphisms in 3’-UTR^7,24,118−120^
*MDM2*, *MDM4*, and *RFWD2*	P53 inhibitors	Increased expression: 1q23.3 gains (*MDM4* and *RFWD2*)^78^
*CDKN2A*	Encoding ARF, an MDM2 inhibitor	Decreased expression: deletions^14,125,126^
*BCL2L12*	P53 inhibitor and caspase 3/7 inhibitor	Increased expression: amplifications^78^
*PERP*	Target of P53, caspase 8 activator	Decreased expression: deletions^78^
*SCOTIN*	Target of P53, pro-caspase 3/7 activator	Decreased expression: deletions^78^
**NF-kB pathway**	Upregulation of several anti-apoptotic proteins from the intrinsic and extrinsic pathways	
*CD79A/CD79B*	Activation of NF-kB pathway via transduction of BCR signaling	Increased expression: gain-of-function mutations in ITAM^14,144,146^
Negative regulators of BCR signaling	Inhibition of NF-kB pathway via downregulation of BCR signaling	Decreased expression: loss-of-function mutations^14^
CARD11-BCL10-MALT1 complex	Activation of NF-kB pathway	Increased expression: gain-of-function mutations in coiled-coiled domain of *CARD11*, chromosomal rearrangements, mutations, and gains of *BCL10*, gains of *MALT1* (18q21)^14,17,147,149−152^
*NFKB2*	Encoding p100, which yields the p52 NF-kB transcription factor	Increased expression: 10q24 rearrangements leading to loss of 3’-end and constitutive protein activation^154^
*REL*	Encoding c-Rel, an NF-kB transcription factor	Increased expression: amplifications^9^
*A20*	Downregulation of NF-kB response	Decreased expression: loss-of-function mutations, 6q23 deletions, mutations in the A20 binding partner *TNIP1*^14,153,156,157^
*MYD88*	Upregulation of NF-kB response via toll and IL-1 signaling	Increased expression: gain-of-function mutations (L265P)^158^
Regulators of NF-kB	Modulation of NF-kB response	Increased expression (positive regulators), decreased expression (negative regulators): mutations^153^

BCR: B cell receptor, UTR: untranslated region, ITAM: immunoreceptor tyrosine-based activation motif, IL-1: interleukin-1.

**Table 3 cancers-13-02167-t003:** Therapies modulating apoptosis being currently investigated clinically in DLBCL.

Investigational Therapy	Mechanism of Apoptosis Modulation	Apoptotic Pathway Affected
**Venetoclax**	BH3 mimetic and BCL2 inhibitor	Intrinsic apoptotic pathway
**CAR T cells**	Genetically modified cytotoxic T cells	Extrinsic apoptotic pathway
**Bispecific T cell engagers**	Linking of B cells and cytotoxic T cells	Extrinsic apoptotic pathway
**Immune checkpoint inhibitors**	Immune checkpoint inhibition, reactivation of cytotoxic T cells	Extrinsic apoptotic pathway
**TRAIL agonists**	Enhancement of the TRAIL/TRAILR response	Extrinsic apoptotic pathway
**Ibrutinib**	Inhibition of BTK, inhibition of BCR signaling	NF-kB pathway
**Lenalidomide**	Immunomodulation, inhibition of BCR signaling	NF-kB pathway
**Bortezomib**	Proteasome inhibitor, decreased degradation of inhibitory kB proteins	NF-kB pathway

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
