# Peer review of "Genetic Events Inhibiting Apoptosis in Diffuse Large B Cell Lymphoma"

_cancers, 2021, doi:10.3390/cancers13092167_

Round 1
Reviewer 1 Report
The current article entitled," Genetic Events Inhibiting Apoptosis in Diffuse Large B-cell Lymphoma” is quite interesting and has scientific merits to be considered for publication.
Authors nicely and comprehensively summarize in this review article that provides a description of the genetic events inhibiting apoptosis in DLBCL, their contribution to lymphomagenesis and chemoresistance, and their implication for the future of DLBCL therapy.
Altogether review article is well written.
Please show schematic diagram of FAS, NF-kB pathway and TRAIL Pathway
What are the different selected drugs targeting aberrant cell survival in diffuse B cell lymphoma.
Summarize different Ongoing trials with drugs targeting cell survival in diffuse large B cell lymphoma.
Reviewer 2 Report
This is a timely review of a large and rapidly evolving field of great relevance to the diagnosis and clinical management of patients with diffuse large B-cell lymphoma (DLBCL), the most common lymphoid malignancy. The manuscript is well organized and clearly written, although it would nonetheless benefit from careful proof-reading to correct fairly numerous, but trivial, grammatical errors and typos. I think that this work will of interest to a reasonably broad readership. My comments that follow, most of which are minor, are intended to be constructive and are offered for the authors’ consideration with the intention of improving the article, perhaps mostly with respect to style and ease of comprehension.
- In my judgment, more care might be taken to adhering consistently to the theme of “genetic events”. The emphasis on genetic events is justified convincingly in the Introduction by referring to the importance of ctDNA, and this idea is laudably revisited at the end. Personally, I find the frequent digression into up- and down-regulation of expression of various players a bit distracting. Some reference to this is important in order to address mechanisms, but care should be taken not to lose track of the general theme that emphasizes mutations (including large ones such as those that are detectably cytogenetically).
- With respect to Table 1, would it possible to avoid having it spread across a page break? It seems to me that the fourth column header might be changed from “...GEP subtype” to “...COO subtype”. Also, would it be possible to add some indication as to what proportion of cases are accounted for by at least some of the subtypes, perhaps the larger ones?
- Figure 1 is very nice. Perhaps indicate on the figure those proteins that constitute the DISC.
- While Table 2 aims to be reasonably comprehensive and was certainly helpful to my understanding, it would benefit from a few edits. For example, although the title refers to “Main genetic events...”, the “Alterations” column includes many references to up- and down-regulation of gene expression, most of which are presumably brought about by trans-acting regulators. These are not genetic alterations; perhaps limit the table’s content to mutated genes? Removing references to transcriptional alterations would make the table smaller and a bit easier for readers to digest. It seems to me that up- and down-regulation is already addressed quite well in the running text; this could be augmented if the authors feel that this is necessary. Adding citations in each row of the table to key articles would be nice. Perhaps consider breaking the content up into more than one table to ease the formatting challenge.
- The authors might consider adding a section on implications for pathological diagnosis and molecular screening. Would they recommend panel sequencing by NGS of all new cases? How about transcriptomics, molecular cytogenetics, etc...? Will conventional classifications systems, such as COO, become obsolete as we get better at directly ascertaining the status of various apoptotic players?
- Care should be taken to ensure that genes are consistently denoted in all caps and italics and that proteins are not (ex., “BCLW” on page 7).
- Venetoclax is introduced on page 7 without a brief explanation as to what it is (i.e.,. a BCL2 antagonist).
- The roles of MEK/ERK and AKT/PI3 kinase signaling downstream of the BCR should at least be mentioned.
- At the top of page 11, phosphorylation and activation of CARD11 are mentioned without explaining what the activity of this protein is (a guanyl kinase and scaffold protein).
- In the discussion of MYD88, the value given for the prevalence of mutations in ABC DLBCL (~30%) seems a bit high. It might also be worthwhile to add that the prevalence of this mutation is, in fact, very high in primary testicular and certain other primary extra-nodal types of DLBCL.
